# Epidemiology of snake envenomation from Mato Grosso do Sul, Brazil

**Karoline Ceron**[1]*, **Cássia Vieira**[1], **Priscila Santos Carvalho**[1,2], **Juan Fernando Cuestas Carrillo**[1], **Jaqueline Alonso**[1], **Diego José Santana**[1]

1 Mapinguari—Laboratório de Sistemática e Biogeografia de Anfíbios e Répteis, Instituto de Biociências, Universidade Federal de Mato Grosso do Sul, Cidade Universitária, Campo Grande, Brazil, 2 Instituto de Biociências, Letras e Ciências Exatas, Universidade Estadual Paulista (UNESP), São José do Rio Preto, Brazil

* adenomera@gmail.com

**Data Availability Statement:** All relevant data are within the manuscript and its Supporting Information files.

**Funding:** KC thanks Fundect (Fundação de Apoio ao Desenvolvimento de Ensino, Ciência e

## Abstract

Snake envenomation is considered a public health problem in tropical countries, where they occur in a high incidence. The present study reports the snake envenomation that occurred in Mato Grosso do Sul state (Brazil) between 2007 and 2017. Epidemiological data were obtained from the online platform of the Notification Disease Information System and were analyzed according to biome. A total of 5568 cases of snake envenomations were recorded during the study period, where the highest frequency was registered between October and April. The majority of envenomations occurred in working-age males (20 to 39 years), caused mainly by *Bothrops* snakes, and the duration of care after the envenomation in most cases took three hours. The municipalities that showed the highest snake envenomations case per 100,000 inhabitants presents low population density, and have their economy based on agricultural activity, which is a risk factor to snake envenomations. To the Mato Grosso do Sul state, the total number of snake envenomations had a positive relationship with the size of the municipality. Since this, larger areas usually have a mosaic of environments, which may harbor higher richness and abundance of snakes, and can cause more snake encounters with the population, resulting in more snake envenomations.

## Author summary

Brazil is the country in South America with the most reported snake envenomations, however, the incidence of snake envenomations is not equal throughout the country. In most cases, the occurrence of snake envenomations is related to environmental factors (e.g., climate) and the increase of human activity in fieldwork. Besides those factors, problems in urban infrastructure, municipality size, or population size are also variables that can influence the number of snake envenomations in a region. The authors found that in the Mato Grosso do Sul state, Brazil, the number of snake envenomations is low when compared with Mato Grosso and Goiás states, and the national average. Despite this, snake envenomations in Mato Grosso do Sul confirmed trends observed in other Brazilian regions, happening mainly in October and April. Most of the envenomations occurred in males by

Tecnologia do Mato Grosso do Sul) for scholarship # 71/700.146/2017. KC, PSC and JFCC thanks Coordenação de Aperfeiçoamento de Pessoal de Nível Superior - Brasil (CAPES) - Funding code 001. DJS thanks CNPq (Conselho Nacional de Desenvolvimento Científico e Tecnológico) for productivity funding (309420/2020-2). The funders had no role in study design, data collection and analysis, decision to publish, or preparation of the manuscript. This study was financed by the Coordenação de. Aperfeiçoamento de Pessoal de Nível Superior - Brasil (CAPES) -. Finance Code 001.

**Competing interests:** The authors have declared that no competing interests exist.

*Bothrops* snakes and with an attendance time of 3 hours. Also, the authors found that there is a relationship between municipality areas with the number of snake envenomations. Seen this, municipalities with larger areas usually have a mosaic of environments, which may harbor higher richness and abundance of snakes, and can cause more snake encounters with the population, resulting in more snake envenomations.

## Introduction

In tropical countries, snake envenomations represent a major problem for public health care due to their morbidity and mortality [1]. In July 2017, the World Health Organization started to recognize snake envenomations as a neglected tropical disease, occurring mainly in rural workers from developing countries in tropical and subtropical regions [2]. Annually, 4.5 to 5.4 million snake envenomations occur worldwide, of this, 1.8 to 2.7 million develop clinical illness, and 81 thousand to 138 thousand die as a consequence of those envenomations. Estimates support that 400 thousand people suffer consequences such as restricted mobility, amputation, blindness, and post-traumatic stress [2].

Brazil is the South American country with the most reported snake envenomations, with close to 28.000 cases per year; the North and Central West areas are at the top of the list [3,4]. Most of the snake envenomations from Brazil are caused by four genus: *Bothrops* (jararaca), *Crotalus* (cascavel), *Lachesis* (surucucu), and *Micrurus* (corais-verdadeiras) [4,5]. Snake envenomations occur with higher frequency at the end and beginning of the year, the major activity time for snakes, and the victims are mainly male rural workers 15 to 49 years old, who get bit mainly on their lower limbs [6,7]. Lethality, in general, is relatively low (0.4%) and depends on different factors such as the time between the envenomation and medical attendance, the identification of the venom type, and later antivenom therapy [4,8].

Several factors are known to interfere in the severity of the snake envenomations. Some are related to the snake, the patients, and the medical assistance, such as which species caused the envenomation or the time between the bite and the administration of the antivenom serum [1]. However, in most cases, the occurrence of snake envenomations are related to environmental factors (e.g., climate, humidity, temperature, and rainfall) and the increase of human activity in fieldwork, mainly during the months of highest rainfall [1]. Besides those factors, problems in urban infrastructure, municipality size, or population size from a rural area, are also variables that can influence the number of snake envenomations in a region [9].

Snake envenomations studies with the proper animal identification are essential. They provide better information to improve medical attendance and treatments, mainly for reducing decision time and antivenom serum application that have direct impacts in reducing local damage, systemic damage, and possible aftermath effects for the victims [10]. Thus, we aim to present the epidemiologic profile of the snake envenomations registered in Mato Grosso do Sul (Central-West Brazil) from 2007 to 2017, and relate the incidences of snake envenomations to local and regional variables.

## Methods

### Study area

We collected data for Mato Grosso do Sul (MS) in the Central-West region of Brazil, an area with 357,125 km$^2$. In 2010, the population was 2,449,024, with 85% of people from urban areas and 15% from rural areas [11]. Mato Grosso do Sul is located in three different biomes,

Cerrado, Pantanal, and Atlantic Forest, harboring 10 species of venomous snakes and 103 species of non-venomous snakes [12,13]. The Cerrado is located in the central-east part of the state, includes most of the state territory, and is one of the most threatened biomes in the world [12]. The Pantanal covers over 25% of the state territory and is located in the western part of the state. Seasonal flooding is one of the main features of this biome [14]. The Atlantic Forest is located in the southern portion of the state and covers 14% of the territory [15]. Atlantic Forest is not only one of the most biodiverse biomes in the world but also provides essential ecosystem services for over 145 million Brazilians living in it [16].

## Data collection

We collected snake envenomations data between 2007 and 2017 in Mato Grosso do Sul state using the Sistema de Informação de Agravos e de Notificações (SINAN) from Ministério da Saúde do Brasil available at http://tabnet.datasus.gov.br/cgi/deftohtm.exe?sinannet/cnv/animaissc.def. SINAN is a platform for the collection and processing of data on notification diseases at the national level, which are notified by health professionals. It contributes to the identification of the epidemiological reality of a given geographical area and allows an analysis of the morbidity profile. Envenomations by animals are considered by the medical staff on a scale from mild to severe. The snake genus that caused the accident and the type of venomous is identified at the initial assessment by the health professionals, according to victims' symptoms.

We collected data such as sex, age group, event date, event location, death occurrence, time between the snake envenomation and medical attendance, numbers of snake envenomations per snake genus, and level of envenomation by state mesoregion. Age groups were: under 1 year old, from 1 to 4, 5 to 9, 10 to 14, 15 to 19, 20 to 39, 40 to 59, 60 to 64, 65 to 69, 70 to 79, and over 80 years old. Envenomation levels were defined as mild, moderate, and severe depending on the symptoms [17]. We collected all data related to each municipality of Mato Grosso do Sul state (population density, biome, and municipality areas) from DATASUS system based on the Brazilian Census 2010 (http://tabnet.datasus.gov.br/).

## Data analyses

We used a variance analysis (ANOVA) to test differences among years. Before ANOVA, we tested normality and homoscedasticity using Kolmogorov-Smirnov test and variance homogeneity with Levene test [18]. After ANOVA we performed Tukey *post hoc* tests to identify differences among years [18]. We assumed the significance level as $p \leq 0.05$.

We used a Chi-Square test to verify differences between males and females. For that, we considered the total number of envenomations as the response variable and sex as the explanatory variable. To verify differences in the occurrence of snake envenomations among months, age groups, and snake species, we used a Kruskal-Wallis non-parametric test followed by a Nemenyi *post-hoc test* [19]. In the first test, we assessed total snake envenomations among months from 2007 to 2017. In the second test, we compared snake envenomations per age group using the following sample sizes: Under 1-year-old, from 1 to 4, 5 to 9, 10 to 14, 15 to 19, 20 to 39, 40 to 59, 60 to 64, 65 to 69, 70 to 79 and over 80 years old. In the third test, we evaluated snake envenomations per snake genus: *Bothrops*, *Crotalus*, *Micrurus*, *Lachesis*, and non-venomous snakes.

We verified the influence of sociodemographic variables (population density, biome, and municipality areas) over the occurrence of snake envenomations with a Generalized Linear Model (GLM) with negative Binomial distribution by using *MASS* [20] package within R program [21]. To this analysis, we removed two leverages from the dataset (Campo Grande and Corumbá municipalities).

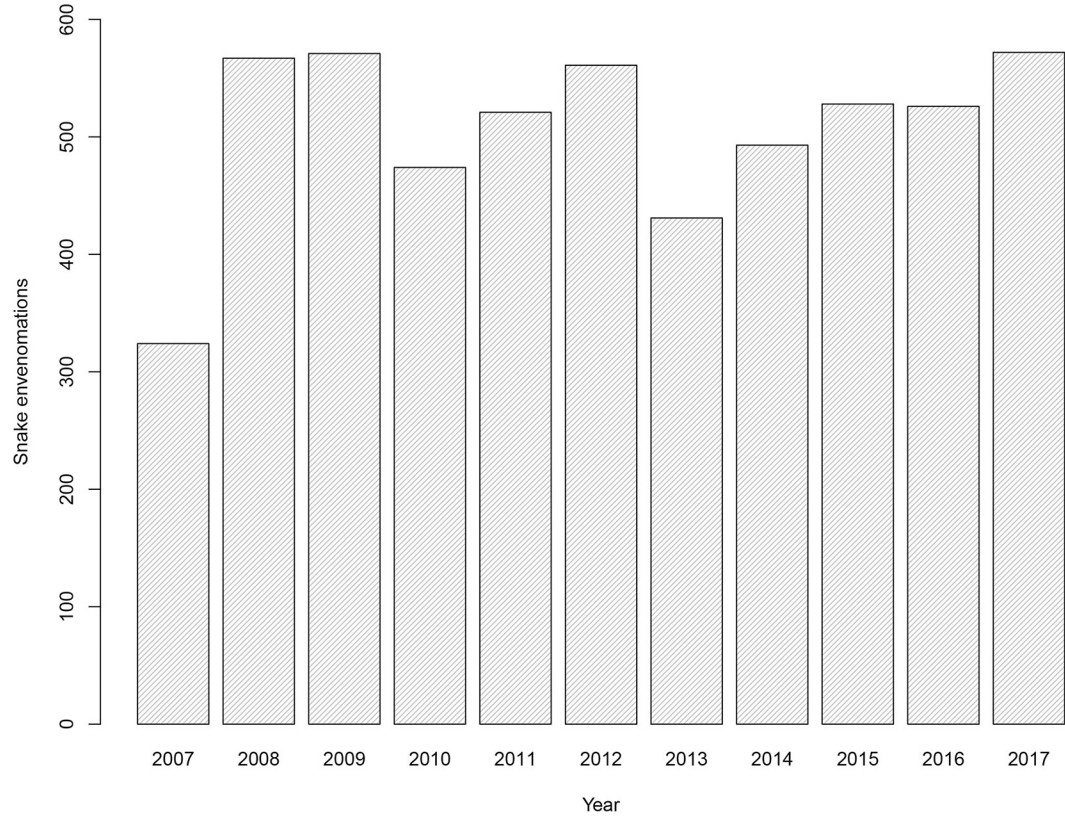

**Fig 1. The number of snake envenomations per year between 2007 to 2017 in Mato Grosso do Sul state (Brazil).**

## Results

We recovered information from a total of 5565 cases of snake envenomations between 2007 and 2017 at Mato Grosso do Sul state (Fig 1). Year 2017 had the most incidents (n = 572) and 2007 had the least incidents (n = 324). The number of snakes envenomations per year differed significantly (F = 2.103, df = 10, p = 0.02), as did snake envenomations per month ($x^2$ = 80.3, df = 11, p < 0.0001). During 2007, April, June, July, August, and September exhibited the fewest cases (p <0.05) (Fig 2). We find a significant difference between sexes ($x^2$ = 1495.1, p ≤0.01), with 75.90% (n = 4226) of the cases occurring in males whereas only 24.08% (n = 1341) occurred in females (Table 1).

We found significant differences among age groups ($x^2$ = 110.31, df = 10, p < 0.0001), with more snakes envenomations occurring in age groups 20 to 39 (35.94%) and 40 to 59 (27.50%). Most of the analyzed cases (73.10%) were attended to within 3 hours of an envenomation and 39.48% within 1 hour (Table 2). We found differences among snake genus ($x^2$ = 50.687, df = 4, p < 0.0001) with 77.07% (n = 4.291) of the cases related to *Bothrops*, followed by *Crotalus durissus* 9.05% (n = 504), *Micrurus* 0.66%, (n = 37), and non-venomous snakes 3.02% (n = 168); six cases were assigned to *Lachesis* (0,11%).

For the Mato Grosso do Sul state and during the studied period, over 15 people died due to envenomations involving snakes corresponding to 0.27% of the total snakes envenomations registered; seven deaths were related to *Bothrops*, three to *Crotalus* and one to *Micrurus* (Fig 3). The rest of the deaths (n = 4) did not have an associated genus. In contrast, most of the people (84.88%, n = 4227) who suffered snake envenomations recovered successfully, one

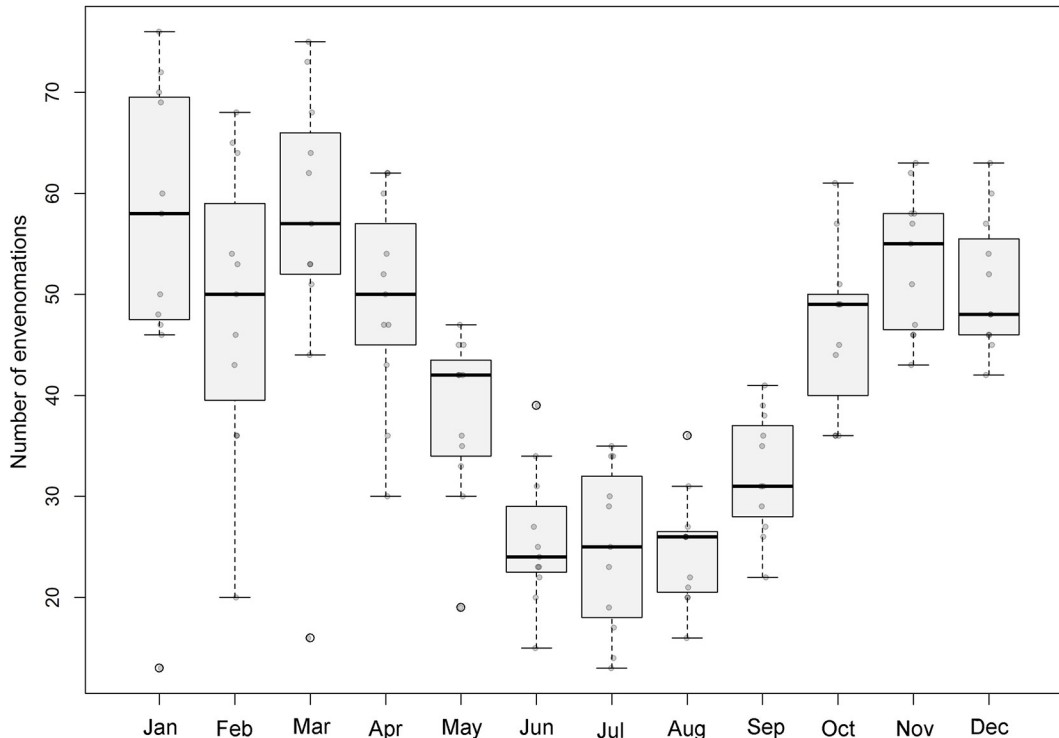

**Fig 2. The number of snake envenomations per month between 2007 to 2017 in Mato Grosso do Sul state (Brazil).**

person died by other causes not related to the snake envenomation (0.01%) and, 826 people (14.83%) did not inform about case evolution. Among the registered cases, 51.02% (n = 2841) were categorised as mild, 33.41% (n = 1860) as moderate, and 8.31% (n = 463) as severe (Table 3).

Based on spatial distribution, most of the envenomations occurred in Cerrado (62.92%), followed by Atlantic Forest (28.81%), and Pantanal with 8.27% (Fig 4). The municipality of Figueirão (n = 103.81), followed by Paranhos (n = 89.29) and Tacuru (n = 82.58) showed the highest incidence of snake envenomations per 100,000 population (Fig 4). However, GLM showed that the number of snake envenomations per municipality in Mato Grosso do Sul state is only related to the municipality area (Z = 4.26, $p \leq 0.01$, Fig 5). The variables of biome and population density did not influence the snake envenomations in Mato Grosso do Sul state ($p > 0.05$).

## Discussion

Snake envenomations epidemiology in Mato Grosso do Sul confirmed trends observed in other Brazilian regions e.g., [4,7,22], happening mainly in October and April. Most of the

**Table 1. Number of snake envenomations by sex in Mato Grosso Do Sul state, Brazil, between 2007 to 2017.**

| Sex | 2007 | 2008 | 2009 | 2010 | 2011 | 2012 | 2013 | 2014 | 2015 | 2016 | 2017 | Total | % |
|---|---|---|---|---|---|---|---|---|---|---|---|---|---|
| Blank/Ignored | - | - | - | - | - | - | 1 | - | - | - | - | 1 | 0.02 |
| Male | 251 | 421 | 435 | 383 | 399 | 422 | 320 | 384 | 398 | 400 | 413 | 4226 | 75.9 |
| Female | 73 | 146 | 136 | 91 | 122 | 139 | 110 | 109 | 130 | 126 | 159 | 1341 | 24.08 |
| Total | 324 | 567 | 571 | 474 | 521 | 561 | 431 | 493 | 528 | 526 | 572 | 5568 | 100 |

**Table 2. The number of snake envenomations per time-lapse between envenomation and attendance in Mato Grosso Do Sul state, Brazil, between 2007 to 2017.**

| Time-Lapse (hours) | Year | | | | | | | | | | | Total | % |
|---|---|---|---|---|---|---|---|---|---|---|---|---|---|
| | 2007 | 2008 | 2009 | 2010 | 2011 | 2012 | 2013 | 2014 | 2015 | 2016 | 2017 | | |
| Blank/Ignored | 19 | 38 | 39 | 21 | 30 | 34 | 33 | 23 | 30 | 31 | 22 | 320 | 5.75% |
| 0 to 1 | 114 | 208 | 223 | 185 | 193 | 230 | 180 | 209 | 214 | 214 | 228 | 2198 | 39.48% |
| 1 to 3 | 113 | 205 | 190 | 161 | 186 | 184 | 139 | 166 | 165 | 157 | 206 | 1872 | 33.62% |
| 3 to 6 | 41 | 55 | 59 | 57 | 63 | 61 | 51 | 54 | 57 | 62 | 63 | 623 | 11.19% |
| 6 to 12 | 11 | 24 | 34 | 22 | 17 | 22 | 14 | 11 | 27 | 27 | 20 | 229 | 4.11% |
| 12 to 24 | 22 | 24 | 17 | 16 | 20 | 11 | 4 | 16 | 25 | 23 | 14 | 192 | 3.45% |
| ≥ 24 | 4 | 13 | 9 | 12 | 12 | 19 | 10 | 14 | 10 | 12 | 19 | 134 | 2.41% |
| Total | 324 | 567 | 571 | 474 | 521 | 561 | 431 | 493 | 528 | 526 | 572 | 5568 | 100.00% |

envenomations occurred in males (> 16 years old) by *Bothrops* snakes and with an attendance time of 3 hours. The number of snake envenomations was also influenced by the municipality size. From 2007 to 2017, health authorities from Mato Grosso do Sul state registered 5,568 cases, which is a relatively low occurrence of cases when compared with Mato Grosso state

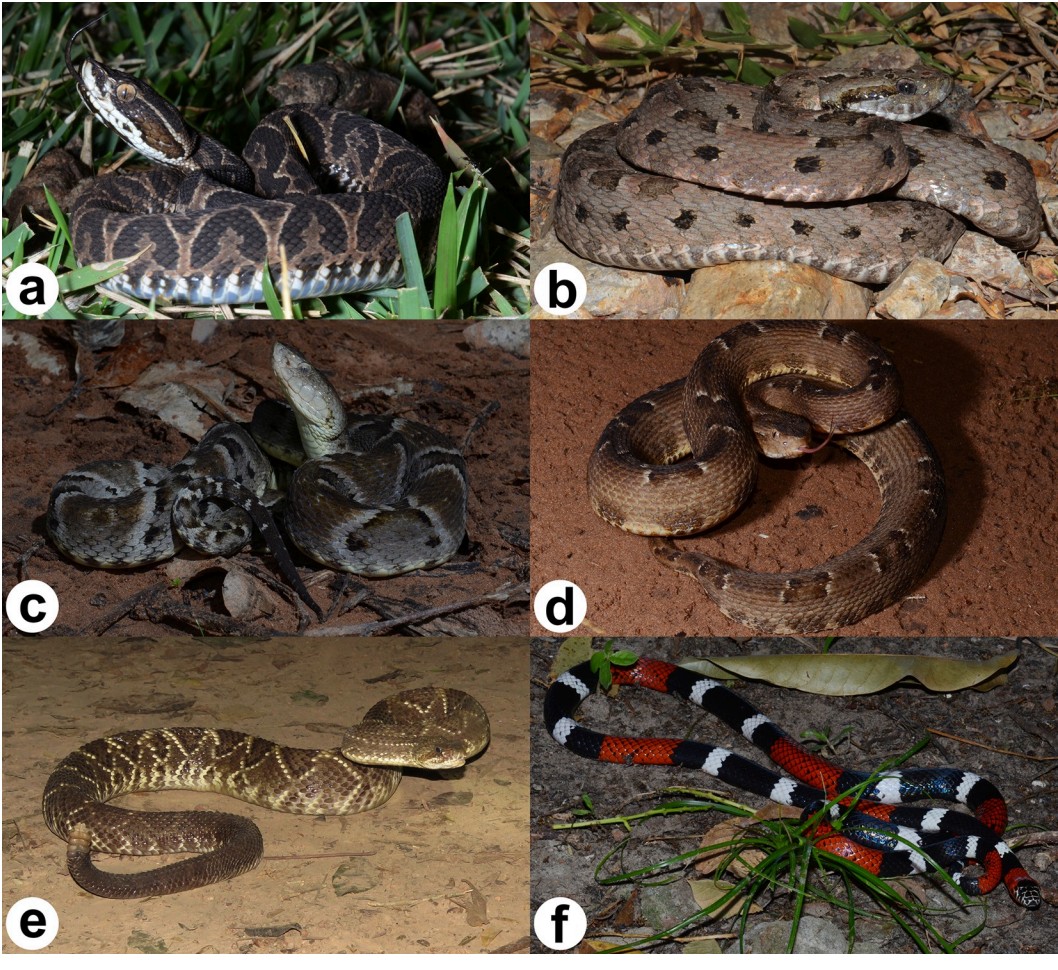

**Fig 3.** Common venomous snakes from Mato Grosso do Sul state (Brazil), where: a) *Bothrops alternatus* (cruzeira), b) *Bothrops mattogrossensis* (boca-de-sapo), c) *Bothrops moojeni* (jararaca), d) *Bothrops pauloensis* (jararaca-pintada), e) *Crotalus durissus* (cascavel) and, f) *Micrurus tricolor* (coral). All photos are from Diego J. Santana, author of the manuscript.

**Table 3. Categorization and number of snake envenomations per snake genus in Mato Grosso Do Sul state, Brazil, between 2007 to 2017.**

| Categorization | Ignored | *Bothrops* | *Crotalus* | *Micrurus* | *Lachesis* | Non-venomous | Total | % |
|---|---|---|---|---|---|---|---|---|
| Ignored | 73 | 277 | 34 | 4 | - | 16 | 404 | 7.26% |
| Mild | 331 | 2108 | 232 | 22 | 4 | 144 | 2841 | 51.02% |
| Moderate | 119 | 1558 | 172 | 4 | 1 | 7 | 1860 | 33.41% |
| Severe | 40 | 348 | 66 | 7 | 1 | 1 | 463 | 8.32% |
| Total | 562 | 4291 | 504 | 37 | 6 | 168 | 5568 | 100.00% |

(13,424), Goiás (11,637), and the national average [23]. Additionally, these data may not reflect the real number of snake envenomations as some envenomations were not reported because they happened in remote rural areas where victims do not have the access to health care [24], as related in other studies carried out in Brazil e.g., [7,10,22]. Besides that, many patients prefer to use alternative medicine rather than a modern treatment provided by a health centre. Unfortunately, this incidence is poorly addressed in Latin America, but it probably plays a significant role in underestimating the incidence and possibly severity of envenomations [25].

There were more snake envenomations notifications in 2017 (n = 572) with the lowest number of records observed in 2007 (n = 324). This low number of reports may be related to the beginning of obtaining data through the online platform, which may cause some data loss and not reflect the actual number of snake envenomations that occurred in that year. The highest number of notifications observed in 2017 could be related to the major use of

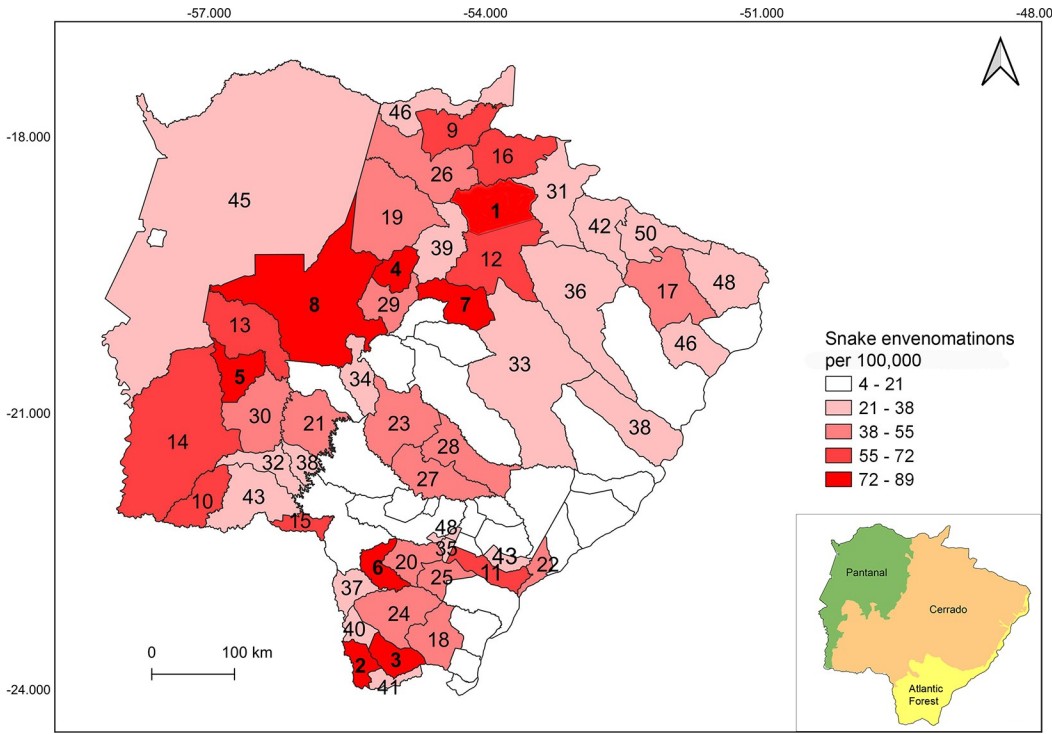

**Fig 4. The mean number of snake envenomation per 100,000 population for municipalities between 2008–2009 and 2011–2017 in Mato Grosso do Sul state (Brazil).** Municipalities codes are listed in S1 Table. IBGE—All maps are in the public domain. (https://www.ibge.gov.br/).

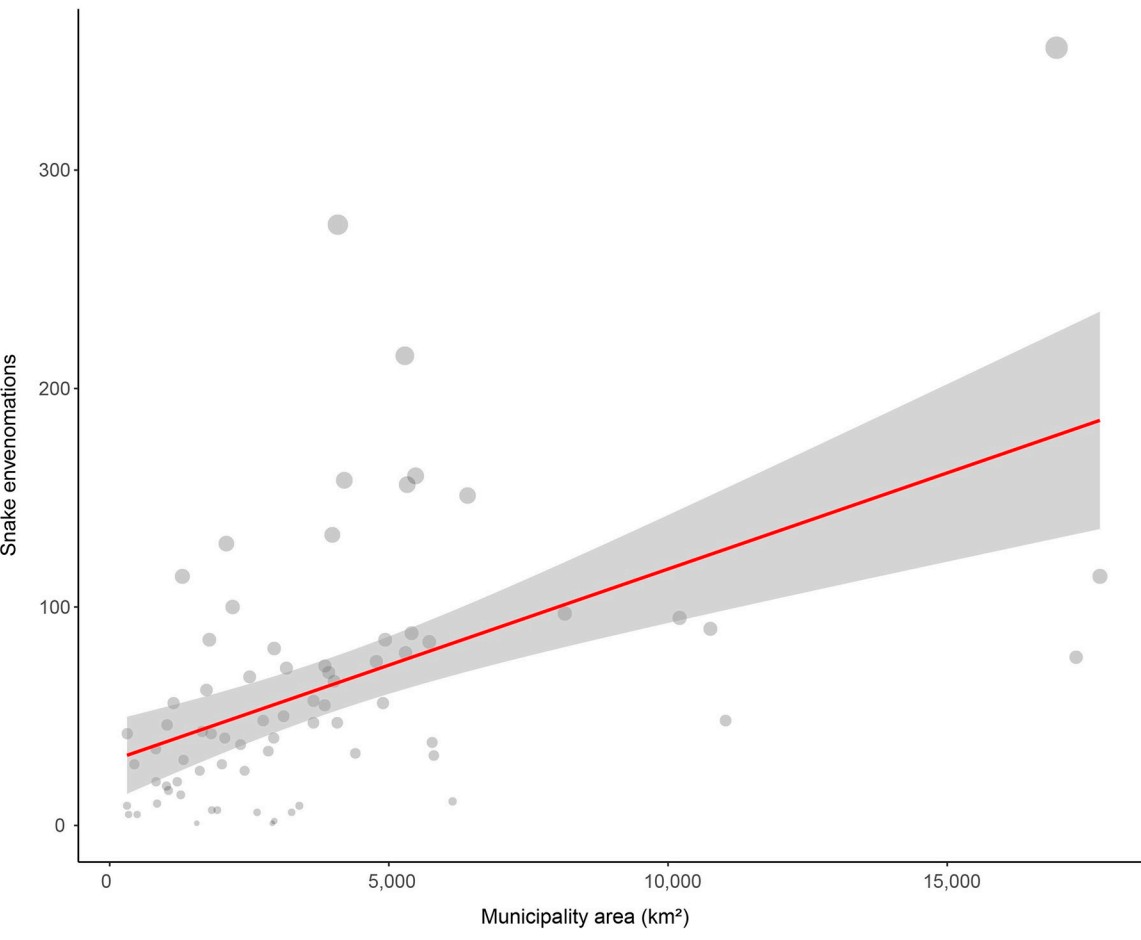

**Fig 5. Relationship between the number of snake envenomation between 2007 and 2017 and the municipality area (Z = 4.26, p ≤ 0.01) of cities from Mato Grosso do Sul state (Brazil).**

the notification platform by the municipalities in the face of modernization of prefectures, allowing internet access to the majority of municipalities resulting in a higher number of envenomations reports. Despite this, it can be related to the growth in the agricultural sector in that year [26], because the agricultural activity itself is a risk factor for the occurrence of snake envenomations worldwide [27]. In 2017, agriculture grew 9.2% in comparison with 2016, with production breaking records in several crops, especially soy (23.9%) and corn (62.9%) [28].

The months between October and April showed the highest number of snake envenomations, which corroborates other studies carried out in Brazil [7,29,30]. This period coincides with the hottest and rainiest season of the year when snakes are most active and prey availability is greater [31]. In addition, the greatest agricultural activity occurs during this time of the year, which increases the number of encounters between rural workers and these animals [4]. The months of June, July, and August had the lowest number of cases, probably due to the lower temperatures during the winter months. Colder and drier periods cause a drop in the metabolism of snakes, which also coincide with the lower availability of prey. Consequently, encounters with these animals are less frequent during these months [31,32].

There was a significant difference between the number of envenomations and the victim's sex. The greater number of snake envenomations in men could be related to the fact that work in the field is performed mainly by men and that agriculture is the main source of income in the state of Mato Grosso do Sul [10,26]. There was a difference in the number of envenomations by age group, with the highest frequency occurring in the age groups of 20 to 39 years old and 40 to 59 years old, both of which involve economically active people. These observations reinforce the idea of envenomations as a work accident because their increase coincides with working-class society. Most of the cases analysed (73.10%) were attended to within 3 hours after the accident and 39.48% were attended within 1 hour after the event. This is probably due to the number of places where snake envenomations can be attended to in Mato Grosso do Sul. From the 76 municipalities in Mato Grosso do Sul state, only 10 do not have centers to attend snakes envenomations. This scenario helps with the prompt attendance of snake envenomations and with the reduction of deaths, once the patient arrives at the medical center; the early stage of treatment is crucial for total recovery. The delay in starting serotherapy can cause local and systemic complications and increase the lethality of envenomations [8]. In addition, with the greater accessibility of the population to media, information regarding public health circulates more quickly and effectively [33], speeding up and improving the treatment of snake envenomations.

There was a difference in the number of cases recorded among snake genus. The genus *Bothrops* was responsible for most of the snake envenomations (77.07%), corroborating the trend observed in other studies in the Brazilian Midwest region [34,35] and other Brazilian states [7,36,37]. Despite this, the majority of cases related to the genus *Bothrops* were classified as mild, according to the symptoms. In Brazil, *Bothrops* cause most snake envenomations, probably due to their abundance, wide geographic distribution, and ability to adapt to different environments [4,38,39]. For the state of Mato Grosso do Sul, *Bothrops moojeni* Hoge, 1966 and *Bothrops mattogrossensis* Amaral, 1925 are the species with the greatest distribution [40,41]. *Bothrops moojeni* (jararaca, caiçaca, baetão) has nocturnal activity and terrestrial habits, generally found in riparian forests of open areas or at forest borders in central and southeastern Brazil [42]. In bothropic envenomation, the venom has proteolytic, coagulant, and haemorrhagic action [43], and the clinical symptoms are mainly local lesions including edema, ecchymosis, pain, hemorrhage, and myonecrosis, associated with systemic changes, such as cardiovascular disorders, hemodynamic injuries, and renal damage [44]. In crotalic envenomation, caused by *Crotalus durissus* (Linnaeus, 1758), the venom has neurotoxic, myotoxic, and coagulant action [43], and the principal systemic evidence are mydriasis, paresthesia, eyelid ptosis, diplopia, generalized myalgia, vomiting, changes in blood pressure, red and brown urine, and acute respiratory failure in severe cases [4,45]. *Crotalus durissus* is a terrestrial snake with crepuscular and nocturnal activity [46]. It inhabits the Cerrado from central Brazil, arid and semiarid regions from the Northeast, and open areas from the south, southeast, and northern Brazil [9]. It can occur in peripheral urban areas but due to its behavior and distribution in these areas, this species causes few cases [34]. Most elapidic envenomation are caused by snakes in the genus *Micrurus.* In Brazil, most of the elapidic envenomation are produced by *Micrurus corallinus* (Merrem, 1820) and *Micrurus frontalis* (Duméril, Bibron and Duméril, 1854) [47]. Only *M. frontalis* has registered in Mato Grosso do Sul [48]. These bites have neurotoxic and myotoxic actions, but injuries are rare because envenomations occur mostly when the victim handles a snake, sometimes children or people under the influence of alcohol [45,49]. This venom induces a slight local reaction, pain and slight edema, paresthesia, and erythema, but the symptoms are mainly neurotoxic, such as eyelid ptosis, ophthalmoplegy (paralysis of the eye muscles), paralysis of the jaws, muscles of the larynx and pharynx, salivation, dizziness, weakness, dysphagia, dyspnoea, and paralysis of the neck and limbs [45,47]. In severe cases, respiratory

failure may occur and end in death [47]. Species in this genus are typically medium-sized snakes, fossorial, not very aggressive, have small venom inoculating teeth, and a limited mouth opening angle [47], all of which may explain the rarity of snake envenomation in this genus.

During the period studied, 0.11% of the envenomations were attributed to *Lachesis muta* (Linnaeus, 1766), known in Brazil as surucucu-pico-de-jaca. This species occurs in the Central West region, only in Mato Grosso and Goiás states, but not in Mato Grosso do Sul [48]. Its venom is proteolytic, coagulant, hemorrhagic, neurotoxic, and the envenomation are similar to bothropic envenomation [50], which may have led to confusion during clinical diagnosis. In addition, more than one species has the same common name leading to confusion and mis-identification or mistakes in SINAN information [51].

In Mato Grosso do Sul during the study period, 15 people died due to snake envenom-ations, representing 0.27% of the total cases registered for the state. Lethality is lower than in states like Mato Grosso (0.52%) and Goiás (0.44%), but higher than in states like Santa Cata-rina (0.24%) and Paraná (0.25%) [23]. Delay or absence of serotherapy favors local and sys-temic complications that directly reflect an increase in the lethality rate of snake envenomations [8]. Early attendance, field worker awareness about serotherapy and serum presence in health units, as well as serum specificity, proper venom administration, and dose, are crucial factors for reducing the lethality of snake envenomations [37,52]. Under the same logic, environmental education can help to reduce envenomations lethality by providing knowledge about snakes, changes in values and improving abilities, basic conditions to stimu-late better integration, and harmony between people and the environment [53]. Myths and leg-ends around snakes interfere with scientific facts, as well as the way films and media deliver misinformation and characterize snakes as cruel animals [54]. Previous studies revealed that the best method to avoid snake envenomations is to deliver prevention measures through the distribution of posters, folders, and lectures [17,38]. In this way, environmental education as an awareness strategy is fundamental for the conservation of snakes by influencing the way people interact with these animals [55], reducing the conflicts between humans and snakes, and reducing the lethality of snake envenomations.

The municipalities that showed the highest snake envenomations case per 100,000 inhabi-tants presents a few populations (from ca. 3000 inhabits to Figueirão to 13000 to Paranhos), which results in low population density, and also have their economy based on agricultural activity [11]. This pattern confirms the trend observed by Chippaux [25] to Bolivia, Argentina, and Colombia, where places with low density, but predominantly agricultural, shown a higher incidence of snake envenomations. The proximity of human populations to the natural envi-ronment explains a greater frequency of encounters with snakes. Consequently, snake enven-omation occurs usually in rural areas during agricultural activities, especially in developing countries, like Brazil, where farming is an important and weakly mechanized economic activ-ity [25].

We did not find a relationship between biome and snake envenomations, which might be related to the generalist habits of some species (e.g., *Bothrops* spp.) that have a wide geographic distribution and high adaptability to different environments [4]. The absence of relation between population density and snake envenomations can be related to the growth of cities, because the human population will be large while human presence may limit the development of snake populations or, even so, snakes will not be able to encounter favorable conditions for their development [56]. On contrary, we found a relationship between municipality area with the number of snake envenomations. In America, snake envenomations are related to snake abundance and the richness of venomous snakes is higher in the tropics [25,57]. Together, municipalities with larger areas usually have a mosaic of environments, which may harbor

higher richness and abundance of snakes, and can cause more snake encounters with the population, resulting in more snake envenomations.

Together, our data highlight the general scenario of snake envenomations in West Brazil and their risk factor involved (working-age males, in bigger municipalities with an economy based on agricultural activity with low population density). In addition, we highlight the urge for more detailed studies involving snake envenomations in West Brazil, clarifying the epidemiologic pattern in focal populations and the clinic evolution of cases.

## Supporting information

**S1 Table. The total and the mean number of snake envenomation per 100,000 population for municipalities between 2008–2009 and 2011–2017 in Mato Grosso do Sul state (Brazil).** (DOCX)

## Acknowledgments

Authors thank Doctor Fritz S. Hertel for English review and comments on the manuscript

## Author Contributions

**Conceptualization:** Karoline Ceron, Priscila Santos Carvalho, Diego José Santana.

**Data curation:** Karoline Ceron, Cássia Vieira, Jaqueline Alonso.

**Formal analysis:** Karoline Ceron.

**Funding acquisition:** Diego José Santana.

**Investigation:** Cássia Vieira, Jaqueline Alonso.

**Methodology:** Karoline Ceron.

**Project administration:** Karoline Ceron, Diego José Santana.

**Software:** Karoline Ceron.

**Supervision:** Karoline Ceron.

**Validation:** Karoline Ceron, Diego José Santana.

**Visualization:** Karoline Ceron.

**Writing – original draft:** Cássia Vieira, Priscila Santos Carvalho, Jaqueline Alonso.

**Writing – review & editing:** Karoline Ceron, Juan Fernando Cuestas Carrillo, Diego José Santana.

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
