## [Decision Letter · Decision Letter 0]

12 May 2021

Dear De Ceron,

Thank you very much for submitting your manuscript "Epidemiology of Snakebites from Mato Grosso do Sul, Brazil" for consideration at PLOS Neglected Tropical Diseases. As with all papers reviewed by the journal, your manuscript was reviewed by members of the editorial board and by several independent reviewers. In light of the reviews (below this email), we would like to invite the resubmission of a significantly-revised version that takes into account the reviewers' comments. 

We cannot make any decision about publication until we have seen the revised manuscript and your response to the reviewers' comments. Your revised manuscript is also likely to be sent to reviewers for further evaluation.

Sincerely,

Arunasalam Pathmeswaran

Associate Editor

Janaka de Silva

Deputy Editor

Reviewer's Responses to Questions

**Key Review Criteria Required for Acceptance?**

**Methods**

-Are the objectives of the study clearly articulated with a clear testable hypothesis stated?

-Is the study design appropriate to address the stated objectives?

-Is the population clearly described and appropriate for the hypothesis being tested?

-Is the sample size sufficient to ensure adequate power to address the hypothesis being tested?

-Were correct statistical analysis used to support conclusions?

-Are there concerns about ethical or regulatory requirements being met?

Reviewer #1: - Objectives: clearly stated 

- Design: clearly exlpained = Retrospective record based study

- Some limitations to address goals: using this design it is difficult to infer any incidence rates (per 100,000 / yr ) as the N of SBE cases who did not consult a health facility are not recorded. This type of study often under-estimates the true incidence. 

Population: Denominator is not stated - who is the population addressed (total district/region population or beyond)? 

- Number of snakebites per snake genus: how was this be assessed in this retrospecive study (at initial assessment, or post-hoc by retrospecive study authors)?- please explain a little more.

- "Elapid accidents were treated as severe..." -> do you mean considered as neurotoxic ? indeed elapids bites can also be dry non-envenomed bites - not all are severe as the quantity of venom injected can be small. A bit unclear. 

- "Lethetic"? I do not undertand this word. Do you mean by Lachesis? 

- Methods lines 114 to 122 : you present Results in the Methods chapter. I think this is more appropriate in the Results chapter.

Reviewer #2: Objective clear. Retrospective analysis should be mentioned on line 89. 

Study area: how many known venomous and non-venomous snake species exist?

Data collection: We need a better description of the national database where the data was collected. Who is collecting it? How is it entered? Who is identifying the snakes? What is the process at which a clinician would enter the data?

We need to distinguish between "snakebite" and "snake envenomation" throughout the manuscript. I have hard time accepting that "poisoning levels" were determined by species and there is no significant clinical data available in this database aside from mortality?

One major question I have is "snakebite" mean "snake envenomation" for data entry.

**Results**

-Does the analysis presented match the analysis plan?

-Are the results clearly and completely presented?

-Are the figures (Tables, Images) of sufficient quality for clarity?

Reviewer #1: - Results : yes , analysis matches well the analysis plan. 

- We usually present the total N of snakebite victims, rather than "snakebites events/accidents". This is because of the possible duplicates. Did any patient consult twice the same year or separate years (ID number identical)? 

- Lines 132-133 , 152-155, etc...: I am not a statistician, but I think that you do not need to present so many details (F value, x2 value and df value): in most studies P-value is sufficient for these simple statistics (You already explained their choice in methods).

- Line 114 (results in methods) compared with Figure 1 : you say "January (n = 6090, February (n =535)"... 

I am surprised by this number for January - is it 6090 or 609 ?? 

- 155: "Differences in snake genera" : It is not clear how this genus was determined : visually by the victim? killed and brought to the clinic/hospital ? extrapolated by the initial clinician ? or by the authors retrospectively? 

- 163: Deaths : it would be useful to know a bit more about these 15 deaths (0.27%). The CFR seems low. Did they occur in children, women? Early deaths (first 24hours) or later ? Syndrome associated with mortality? (rather than species which can be often mistaken). 

Recovery : Most people (84.88%) recovered successfully... What about the 15% who did not recover successfully? - amputations? sequelae? long-term disability? Lost to follow-up % ?

- 169 : grave : I think you mean severe (severe in table 3). 

- How was the categorisation between mild, mod, severe - which criteria ? (Brazilian guidelines? WHO? Other?) 

- 174-178: There is no comparison between number of cases and population (incidence per 100k pop/y) of each geographical area. This would be very important (essential ) to use this type of study, for any epidemiological comparison with studies from other parts of the world. 

- Fig3 is interesting showing N of cases per area and showing a superior number in the Pantanal biome areas; BUT for epidemiologists this figure should be based on numbers per 100,000 inhabitants - this would allow comparisons and would possibly show different colour gradients based depending on pop density. 

-> You could use your total population (2,449,024 in 2010) or rural population (15% only?) as a denominator. Does the percnt of rural pop (< or <15%) vary among the districts in your map. This is important as it can bias your interpretation. Therefore the total population may be more reliable. 

- Fig 4 (b): I am not sure I understand what you mean by "rural population" on the x-axis ? Is this the total population minus the urban population ? I think it would be clearer on the map (Like map Fig 3).

Reviewer #2: Table 3: how did we determine severity of the envenomation? What does ignored mean? Does that mean snake was not identified?

Is there any clinical data available to report? Appears limited

Do we have information on the circumstances of the "snakebites" like occupation and other factors placing increased for a snake encounter?

"non-poisonous" snakes bites were 3.02% and I would consider that low for such a large area. Is this because the initial thought was it was a venomous snake and it was later deemed non-venomous encounter? To only have 168 non-venomous snake encounters over such a large period indicates that the individual entering the data must have thought it was venomous but it was later determined not to be a venomous encounter. We need clarification. 

Among the 15 who died, do we have any more information on the cases? Delay in antivenom or health care?

I would remove the work "accidents" from the manuscript and use envenomation. 

I would consider a figure of the common genera that are found in this region associated with snake envenomation.

**Conclusions**

-Are the conclusions supported by the data presented?

-Are the limitations of analysis clearly described?

-Do the authors discuss how these data can be helpful to advance our understanding of the topic under study?

-Is public health relevance addressed?

Reviewer #1: - Yes but not entirely : you do not really report incidence rates, nor comparative incidence rates. N-cases cannot really be compared if there is no denominator (total population). 

- Lines 205-207- "these data may not reflect the real number of snakebites as some

accidents were not reported because they happened in remote rural areas where victims do not

have the access to health care." -> I agree with your statement BUT: would it be possible to estimate (from other studies) the proportion of victims (incl women) who only consult basic health posts or traditional healers in remote areas? In other countries and all continents , this is our major issue leading to under-estimation (Costa Rica, India, Nepal, Sri Lanka, Cameroon, etc.). It would be interesting to conduct a random multicluster cross-sectional study in the same area to assess this difference in incidence. 

- Your hypothesis on agricultural growth seems interesting but it is not supported by data from the Results chapter (not really a conclusion). 

- I have doubts about this conclusion: "There was no significant difference between the number of accidents and the victim’s

sex even though 75.90% occurred in men and only 24.08% in women." - If you take the total N F and M in the total population (50 & 50) this would give you a very significant difference in a simple 2x2 table (X-square , p-value<0.00001). 

- Same question on age groups: 20-29 and 30-39. P-value comparing age groups? e.g. severe vs mild-mod? In many studies children are significantly more severe, and mortality is higher in children (also in women in some areas in Asia).

Reviewer #2: No limitations are present and there are several. Including data collection and data entry system, which is largely the biggest limitation. We need more discussion on the process of data entry. Is every suspected snake envenomation required to be entered into the system? Who determined the snake envenomation and severity of illness?

**Editorial and Data Presentation Modifications?**

Reviewer #1: -Annually, 1,8 to 2,7 million snakebites 37 occur worldwide and 81 thousand to 138 thousand deaths

 Actually WHO says : 4.5 to 5.4 million bites / 1.8 to 2.7 million snakebite ENVENOMING cases (SBE)... 81-138 thousand deaths.. https://www.who.int/health-topics/snakebite#tab=tab_1 : "WHO...available data show 4.5–5.4 million people get bitten by snakes annually. Of this, 1.8–2.7 million develop clinical illness and 81 000 to 138 000 die from complications."

- Line 46 : hind limbs (usually for animals) -> lower limbs (for humans)

- Line 61: epidemiologic profile of the snakebites : Do you mean present clinical profiles ... and relate them to epidemiological factors-variables? 

- Lines 72 to 85: not sure these details on biodiversity threats are relevant for the analysis. 

- Abstract, last line: "The number of snakebite accidents had a positive relationship with the size of the

 municipality and the population size living in rural areas." it is not clear if this is an association with population size" or with the rural vs urban environment?

Reviewer #2: (No Response)

**Summary and General Comments**

Reviewer #1: Thank you for the opportunity of reading and reviewing this very important piece of evidence about Snakebite Envenoming in Mato Grosso Do Sul, Brazil. 

However, it leaves the reader and probably some snakebite specialists with a little sense that it is unfinished, and not completely focused 

A/ on epidemiology (incidence rates, comparisons, stratifications, environmental risk factors) 

B/ or demographic/clinical risk factors outcomes (severity criteria, snake genus,... ). 

Your paper would benefit by adding : 

1) a table with demographics (age, sex, profession, area, hospital...) of victims

2) a table with main outcomes : severity, deaths, compared to non-severe -> Odds ratios of dying? of being severe ? (by Month/Season? by Year ? by hospital (qlty of care)? 

3) a map with estimated incidence rates per 100k pop/yr using the official census total population (it's ok if you explain the possible bias due to not having community data). 

Looking forward to seeing the next version ! 

Many thanks for this opportunity and best wishes.

Reviewer #2: This manuscript needs some attention but study has validity. I would not use the terms "snakebite" "accident" or "snake accident" in this manuscript. This terminology is inaccurate and not attractive for those in the field. Correct terminology is snake envenomation and this needs to be replaced throughout the manuscript. Many people are bitten by snakes and we need to accurate on the terminology used. I also suggest that using the term "poisoning" is also not accurate when discussing envenomation. Line 96: "poisoning levels" what does that mean? I would also refer to "non-poisonous" snake as "non-venomous" as this is accurate and replace throughout manuscript. Remember, venom is a poison, but a poison is typically referred to when a toxin is ingested, inhaled, or absorbed through the skin. Venom is injected directly into the body via "bite or sting" from an organism.

PLOS authors have the option to publish the peer review history of their article (what does this mean?). If published, this will include your full peer review and any attached files.

Reviewer #1: Yes: Gabriel Alcoba

Reviewer #2: No
---

## [Decision Letter · Decision Letter 1]

6 Jul 2021

Dear Dr Ceron,

Thank you very much for submitting your revised manuscript "Epidemiology of snake envenomation from Mato Grosso do Sul, Brazil" for consideration at PLOS Neglected Tropical Diseases. As with all papers reviewed by the journal, your manuscript was reviewed by members of the editorial board and by several independent reviewers. The reviewers appreciated the attention to an important topic. Based on the reviews, we are likely to accept this manuscript for publication, providing that you modify the manuscript according to the review recommendations. 

Sincerely,

Arunasalam Pathmeswaran

Associate Editor

Janaka de Silva

Deputy Editor

Reviewer's Responses to Questions

**Key Review Criteria Required for Acceptance?**

**Methods**

-Are the objectives of the study clearly articulated with a clear testable hypothesis stated?

-Is the study design appropriate to address the stated objectives?

-Is the population clearly described and appropriate for the hypothesis being tested?

-Is the sample size sufficient to ensure adequate power to address the hypothesis being tested?

-Were correct statistical analysis used to support conclusions?

-Are there concerns about ethical or regulatory requirements being met?

Reviewer #1: Thanks for submiting this revised version answering very well to all my questions-suggestions, especially on the epidemiological parts (per 100k, map, etc).

Reviewer #2: Revision has markedly been improved and methods are clearer to the reviewer.

Study design has validity. SINAN system explained in more detail. 

Need to include in the methods section the explanation on severity scale as it pertains to the FUNASA, 2001. For those outside of Brazil, this method of grading clinical snake envenomation may be different. A statement, like provided in the response letter should be added to lines 127-130 and also referenced.

**Results**

-Does the analysis presented match the analysis plan?

-Are the results clearly and completely presented?

-Are the figures (Tables, Images) of sufficient quality for clarity?

Reviewer #1: Looking at your responses to points 3 and 4, I think it is still unlear how you classified elapid bites (eg by Micrurus/coral) when you look at table 3, in which you classify most bites by Micrurus as mild (22) and moderate (4) and only 7 as severe. 

This does not fit your answer about coral snakebites considered as "severe" (Point 4 below): elapid accidents were treated as severe... it contradicts the classification of Micrurus bites into mild-moderate-severe. I think it's better that you stick to the initial clinical classification as you have no means to verify retrospectively. So I would remove your extrapolation (also the corrected one) saying that "Elapidic accidents with clinical manifestations were

treated as severe" (lines 96-99). 

I do not understand how reliable this table is as it does not seem to be associated with final outcomes (death, surgery, amputations, disability, hospitalisation time). 

**This does not correspond to your answers : 

3. Number of snakebites per snake genus: how was this be assessed in this retrospecive study 

(at initial assessment, or post-hoc by retrospecive study authors)?- please explain a little more.

Response: Envenomations by animals are considered by the medical team as moderate or

severe, and they are the ones who identify which species caused the accident and the type of

venomous, according to victims’ symptoms. We added a sentence in methods explaining

data obtention by SINAN (lines 85-94). Thus, the genus is assessed at initial assessment by

health professionals.

4. "Elapid accidents were treated as severe..." -> do you mean considered as neurotoxic ? 

indeed elapids bites can also be dry non-envenomed bites - not all are severe as the quantity of 

venom injected can be small. A bit unclear.

Response: According to FUNASA, 2001 in the guide “Manual for the diagnosis and

treatment of accidents involving poisonous animals” all cases of coral accidents with clinical

manifestations should be considered as potentially serious. We modified this sentence to be

cleaner, now it reads (lines 96-99): “Elapidic accidents with clinical manifestations were

treated as severe, lethetic accidents were treated as moderate to severe depending on the

symptoms, and botropic and crotalics varied among mild, moderate, and severe [19].”

**

Another point is your answer about having a map with N cases per 100k? I do not find it : the map still shows absolute number of cases, but it does not show Cases/100k pop./year -> It would be great to modify the red map and the colour scale of reds to reflect the real relative incidence rates/100k pop. 

**

In the limitations you should talk about the usual under-estimation due to cases and deaths ocurring outside any medical facility. You would need a random cluster household survey - the next step ?

Reviewer #2: Analysis is sufficient for data extracted from the SINAN database. 

In the response letter the authors have acknowledged limitations and this needs to be added to the discussion section. 

The analysis is descriptive and based-off data entered by health professionals into the SINAN database. Clinical data is not provided and details of the envenomation are lacking. A paragraph of limitations should be added and would add more rigor to their study. It also is a chance to introduce a statement for the need for more detailed study on snake envenomation in this region of Brazil.

**Conclusions**

-Are the conclusions supported by the data presented?

-Are the limitations of analysis clearly described?

-Do the authors discuss how these data can be helpful to advance our understanding of the topic under study?

-Is public health relevance addressed?

Reviewer #1: (No Response)

Reviewer #2: Limitations are not clearly included in the discussion. Public health relevance has been stated.

**Editorial and Data Presentation Modifications?**

Reviewer #1: (No Response)

Reviewer #2: (No Response)

**Summary and General Comments**

Reviewer #1: Besides these 2 final comments, this is a very interesting and useful paper which calls for population-based epidemiological studies, such as multi-cluster cross-sectional surveys... 

Thanks!

Reviewer #2: Overall the manuscript has been significantly improved. 

Two main comments on the revised manuscript:

- the term accidents is scattered throughout the manuscript. In my opinion, "accidents" should be replaced with envenomation. It appears that some studies coming out of Brazil are using the term "accident" for envenomation. The more common terminology should be used and envenomation is more exact. 

- limitations are not clearly discussed in the discussion section. Others from Brazil have been published using data in the SINAN database and this should also be stated.

PLOS authors have the option to publish the peer review history of their article (what does this mean?). If published, this will include your full peer review and any attached files.

Reviewer #1: No

Reviewer #2: No

Figure Files:

Data Requirements:

Reproducibility:

References

---

## [Editor Report · Decision Letter 2]

17 Aug 2021

Dear Dr Ceron,

We are pleased to inform you that your manuscript 'Epidemiology of snake envenomation from Mato Grosso do Sul, Brazil' has been provisionally accepted for publication in PLOS Neglected Tropical Diseases.

Best regards,

Arunasalam Pathmeswaran

Associate Editor

Janaka de Silva

Deputy Editor

---

## [Editor Report · Acceptance letter]

3 Sep 2021

Dear Dr. Ceron,

We are delighted to inform you that your manuscript, "Epidemiology of snake envenomation from Mato Grosso do Sul, Brazil," has been formally accepted for publication in PLOS Neglected Tropical Diseases.

Best regards,

Shaden Kamhawi

co-Editor-in-Chief

Paul Brindley

co-Editor-in-Chief
